# Endoscopic versus Surgical Intervention for Painful Obstructive Chronic Pancreatitis: A Systematic Review and Meta-Analysis

**DOI:** 10.3390/jcm10122636

**Published:** 2021-06-15

**Authors:** Ka Wing Ma, Hoonsub So, Euisoo Shin, Janice Hoi Man Mok, Kim Ho Kam Yuen, Tan To Cheung, Do Hyun Park

**Affiliations:** 1Department of Surgery, The University of Hong Kong, Hong Kong, China; kawingma@hku.hk (K.W.M.); kimyuen1@hku.hk (K.H.K.Y.); cheung68@hku.hk (T.T.C.); 2Department of Internal Medicine, Ulsan University Hospital, University of Ulsan College of Medicine, Ulsan 44033, Korea; Hoon3112@gmail.com; 3Asan Medical Library, University of Ulsan College of Medicine, Seoul 05505, Korea; Euisso@amc.seoul.kr; 4Li Ka Shing Faculty of Medicine, The University of Hong Kong, Hong Kong, China; Janicehm@connect.hku.hk; 5Division of Gastroenterology, Department of Internal Medicine, University of Ulsan College of Medicine, Asan Medical Center, Seoul 05505, Korea

**Keywords:** pancreatitis, systemic review, meta-analysis

## Abstract

There is limited evidence on the standard care for painful obstructive chronic pancreatitis (CP), while comparisons of endoscopic and surgical modes for pain relief have yielded conflicting results from small sample sizes. We aimed to obtain a clear picture of the matter by a meta-analysis of these results. We searched the Pubmed, Embase, and Cochrane Library databases to identify studies comparing endoscopic and surgical treatments for painful obstructive CP. Pooled effects were calculated by the random effect model. Primary outcomes were overall pain relief (complete and partial), and secondary outcomes were complete and partial pain relief, complication rate, hospitalization duration, and endocrine insufficiency. Seven studies with 570 patients were included in the final analysis. Surgical drainage was associated with superior overall pain relief [OR 0.33, 95% CI 0.23–0.47, *p* < 0.001, I^2^ = 4%] and lesser incidence of endocrine insufficiency [OR 2.10, 95% CI 1.20–3.67, *p* = 0.01, I^2^ = 0%], but no significant difference in the subgroup of complete [OR 0.57, 95% CI 0.32–1.01, *p* = 0.054, I^2^ = 0%] or partial [OR 0.67, 95% CI 0.37–1.22, *p* = 0.19, I^2^ = 0%] pain relief, complication rates [OR 1.00, 95% CI 0.41–2.46, *p* = 0.99, I^2^ = 49%], and hospital stay [OR −0.54, 95% CI −1.23–0.15, *p* = 0.13, I^2^ = 87%] was found. Surgery is associated with significantly better overall pain relief and lesser endocrine insufficiency in patients with painful obstructive CP. However, considering the invasiveness of surgery, no significant differences in complete or partial pain relief, and heterogeneity of a few parameters between two groups, endoscopic drainage may be firstly performed and surgical drainage may be considered when endoscopic drainage fails.

## 1. Introduction

Chronic pancreatitis is a prevailing health topic in the western countries, with a reported prevalence of around 50/100,000 persons [1,2]. Contributed by increasing societal affluence, alcohol consumption, and availability of diagnostic imaging, this condition is becoming more common also in the developing countries, ranging from 13.5 to 125 per 100,000 persons [3,4]. Alcohol is the single most common risk factor for chronic pancreatitis [1,5,6], and it predominantly affects men aged 40–60 years, imposing substantial socio-economic burdens. In the United Kingdom, it has been estimated that the direct and indirect costs relating to chronic pancreatitis totaled GBP 285.3 million per year [7]. Apart from alcoholic pancreatitis, autoimmune, metabolic, toxic, hereditary, and idiopathic pancreatitis constitute to the remaining number of the patient, and their symptoms typically recur despite medications or lifestyle modification. Abdominal pain is a leading cause of hospitalization in patients with chronic pancreatitis. Data from North American Pancreatitis Study 2 Continuation and Validation, a prospective multi-center study, showed that 66.8% of the patients experienced severe abdominal pain [8]. Such pain is commonly a result of pancreatic ductal obstruction secondary to stricture and stone formation although repetitive parenchymal inflammation also plays a major role in some non-obstructive cases. Medical treatments such as opioid-based analgesics or drugs that modulate neuropathic pain are effective for short term pain suppression, while more lasting pain control requires adequate pancreatic ductal drainage, which is chiefly done by endoscopic or surgical approach.

Stepwise escalation of treatment aggressiveness has been advocated [9,10], starting from oral analgesic regimens [11], followed by less invasive endoscopic drainage with or without extracorporeal shockwave lithotripsy for painful obstructive chronic pancreatitis. If these measures are deemed unsuccessful, surgery will be contemplated as the last resort [12,13]. The upside of this approach is that major surgery is avoided when the endoscopic treatment succeeds. A large multi-center retrospective study reported that endoscopic treatment resulted in long-term pain improvement in 80% of the patients [14]. However, a couple of studies reported that a significant proportion of patients remained in significant pain after a period of endoscopic treatment [15,16] and eventually needed a surgical procedure. In the literature, there are retrospective studies [17,18,19,20] and prospective randomized controlled trials (RCTs) [21,22,23] carried out to investigate the efficacy of different approaches to pain control in chronic pancreatitis. Moreover, recent RCT showed early surgery had lower pain scores compared with endoscopy-first approach [21]. In view of their conflicting results and relatively small sample sizes, we performed a systematic review and meta-analysis to look into this important topic.

## 2. Materials and Methods

### 2.1. Search Method

The current meta-analysis has been registered in the National Institute of Health Research (PROSPERO registration number: CRD42020207014).

The systematic review followed the Preferred Reporting Items for Systematic Reviews and Meta-Analyses (PRISMA) [24] and Meta-analysis Of Observational Studies in Epidemiology (MOOSE) [25] guidelines. Literature review was performed according to a search protocol agreed by all authors. The statistical teams at Asan Medical Center and the University of Hong Kong, respectively, carried out independent literature search on different search engines, namely Pubmed, EMbase (embase.com), and Cochrane Library (Wiley) [26]. MESH terms used included “pancreatitis, chronic”, “chronic disease AND pancreatitis”, “pain”, “endoscopy”, and “surgical procedure”. The period for search was from inception to 12 May 2020. Studies that used combined shockwave lithotripsy and endoscopy and studies published in non-English languages were excluded. Details of the literature search are shown in the supplementary file. In case of insufficient data or data requiring clarification, the corresponding authors of the particular studies were contacted by email.

### 2.2. Outcome Assessment

For the purpose of this meta-analysis, primary outcomes were all levels of pain relief and secondary outcomes were proportions of complete and partial responses, complication rate, hospitalization duration, and endocrine and exocrine insufficiency. For the primary outcomes, pooled rate of overall pain relief is defined as the total proportion of complete and partial pain relief. The pooled rate of complete pain relief corresponds to the composite of Izbicki pain score ≤10, or the absence of painful attacks. At the same time, pooled rate of partial pain relief corresponds to the composite of Izbicki pain score >10 with a more than 50% decrease compared with the baseline score, or a reduction in pain of at least three points on the Melzack score. For the secondary outcomes, the pooled complication rate is defined as endoscopic complications, including pancreatitis, bleeding, stent migration, and surgical complications, including anastomotic leakage and pancreatic fistula. The pooled endocrine and exocrine insufficiency rate is defined as the use of diabetes medication, and as composite of fecal elastase <200 μg/g, pancreatic functioning diagnosant <70%, or new-onset steatorrhea, respectively.

### 2.3. Statistical Analysis, Assessment of Publication Bias and Sensitivity Analysis

Meta-analysis was performed using R Software, version 4.0.1. Since heterogeneity across the included studies was expected, the random effect model was used instead of the fix model analysis. I^2^ test result with number less than 25% was regarded as absence of significant heterogeneity. Independent samples test was applied to compare the means of two groups. Funnel plot with Eggar’s test was used to check for publication bias. *p* values of less than 0.05 were considered statistically significant. Sensitivity analysis was conducted for primary outcome to delineate the accuracy of the pooled results using the receiver-operating characteristics curves.

### 2.4. Assessment of the Quality of Individual Studies

The quality of the three RCTs was assessed with Jadad scoring, which rates a study according to three main criteria: 1, randomization; 2, blinding; and 3, withdrawals and dropouts [27]. The Jadad scale ranges from 0 to 5. Studies with a score of 3 or above were deemed satisfactory. The Newcastle–Ottawa scale (NOS) was applied to the four retrospective studies, with importance placed on sample selection, comparability, and outcome [28]. Scores range from three to nine, and studies of satisfactory quality commonly score six or above. The assessment was performed by two authors independently. Any disagreement on the scoring was resolved by the corresponding author.

## 3. Results

The literature search protocol identified 2616 potential studies (Figure 1). After exclusion of duplicates and studies that did not report direct comparison of endoscopic and surgical treatments for chronic pancreatitis, seven studies—four retrospective cohort analyses [17,18,19,20] and three RCTs [21,22,23]—remained (Table 1). In total, 570 patients were included for analysis. The majority (72.1%) of them were male and more than a half (53.3%) had alcoholic pancreatitis. The mean age was 50.5 years. The average follow-up period was 50.7 months.

### 3.1. Pain Relief after Endoscopic and Surgical Treatment

As to pain relief assessment, three studies [19,21,22] used the Izbicki pain score [29], one [23] used Melzack score [30], one [17] used reduction in dosage, and two [18,20] did not report their methods of pain relief assessment. Three studies [17,18,22] found no difference in pain relief between the two modalities, while four studies [19,20,21,23] reported superior pain relief with the surgical approach. Our meta-analysis of these seven studies demonstrated that surgical drainage was associated with better overall pain relief (complete and partial) as the primary outcome [OR 0.33, 95% CI 0.23–0.47, *p* < 0.001, I^2^ = 4%] (Figure 2). Four studies [18,21,22,23] reported both complete pain relief (Figure 3) and partial pain relief (Figure 4) as the secondary outcome. Although statistical difference was not demonstrable between the two treatment approaches regarding complete [OR 0.57, 95% CI 0.32–1.01, *p* = 0.054, I^2^ = 0%] and partial [OR 0.67, 95% CI 0.37–1.22, *p* = 0.19, I^2^ = 0%] pain relief, it was noted that surgical drainage tends to have a higher rate of complete pain relief (Figure 3).

### 3.2. Other Treatment Outcomes

#### 3.2.1. Hospital Stay, Procedure-Related Complications, and Mortality

The length of hospital stay was reported in five studies [17,18,19,21,22], with a tendency of shorter stay in the endoscopic group. The median period of stay was 28.4 days in the endoscopic group and 36.8 days in the surgical group. Four [17,19,21,22] out of the five studies reported shorter stay in the endoscopic group, while one [18] reported that the total mean stay was longer in the endoscopic group, which had more hospital admissions. The single mean hospital stay was significantly shorter in the endoscopic group. Our meta-analysis, however, found no significant difference in length of hospital stay between the two groups [OR −0.54, 95% CI −1.23–0.15, *p* = 0.13, I^2^ = 87%] (Figure 5).

Rates of complication and mortality were reported in five studies [17,18,19,21,22]. The procedural mortality rate was 1.2% in the endoscopic group and 0.6% in the surgical group. No statistically significant difference in the occurrence of overall complication between the two groups was observed [OR 1.00, 95% CI 0.41–2.46, *p* = 0.99, I^2^ = 49%] (Figure 6).

#### 3.2.2. Endocrine and Exocrine Insufficiency

Five papers [17,18,19,21,22] provided comparative data regarding endocrine insufficiency. It was noted that different definitions of endocrine insufficiency were adopted. Four studies [18,19,21,22] defined it as a new onset of diabetes mellitus or the need for glycemic control, whereas the other study [17] used the increase in HbA1c level > 6.1% as the definition. Despite the heterogeneity in the definitions adopted, all these studies reported superior outcomes with the surgical approach. The overall incidence of endocrine insufficiency was 29.8% in the endoscopic drainage group versus 20.0% in the surgical drainage group. The difference was statistically significant [OR 2.10, 95% CI 1.20–3.67, *p* = 0.01, I^2^ = 0%] (Figure 7).

Six papers [17,18,19,20,21,22] reported data regarding exocrine insufficiency; similarly, different definitions were adopted. Exocrine insufficiency was defined as fecal elastase <200 μg/g in two studies [21,22], as new onset of steatorrhea in another [18], and as pancreatic functioning diagnosant level <70% in yet another [17]. The other two studies [19,20] did not report the assessment method for exocrine function. Meta-analysis was not possible due to the gross inconsistency of definitions. Nonetheless, these papers demonstrated slightly superior outcomes in the surgical group (Table 1). The overall incidence of exocrine insufficiency was 54.5% after endoscopy and 44.5% after surgery. The difference was, however, not statistically significant [*p* = 0.46].

### 3.3. Qualitative Assessment of the Included Studies

The three RCTs [21,22,23] had a mean Jadad score [27] of 2.67 (range 2–3), indicating medium quality (Table 2). Their methods of randomization were suitable and clearly defined. Two [21,22] out of the three papers reported withdrawal and dropout rates. However, treatment involving endoscopy and surgery made blinding impossible, which limited the quality of the three studies.

The four retrospective cohort studies [17,18,19,20] had a mean NOS of 7.25 (range 6–9) (Table 2). The overall quality of the studies was satisfactory. Patient selection and treatment outcomes were clearly documented. Nonetheless, the study populations were not fully comparable due to treatment preferences concerning patients’ clinical situations. A brief follow-up period and a high dropout rate also limited the quality of a couple of the studies [17,23].

### 3.4. Assessment of Publication Bias of the Included Studies

Regarding the effect of drainage approach on pain relief (partial or complete), results from four studies [19,20,22,23] favored surgical drainage while those from three studies [17,18,21] favored endoscopic drainage. The funnel plot demonstrated an even distribution of the seven studies, suggesting insignificant publication bias [Eggar’s test *p* = 0.40] (Figure 8). Sensitivity analysis was performed to test the validity of I-square value in the pooled result (random effect model) of primary outcome. The optimal specificity and sensitivity (in the Youden index sense) for summary ROC curve are 0.8 and 0.523, respectively, resulting in a value of 0.677 for the area under the summary ROC curve, signifying consistent heterogeneity with the I-square test value (Figure 9).

## 4. Discussion

The current meta-analysis demonstrated that surgical drainage is associated with better overall pain relief for patients with chronic pancreatitis in comparison to endoscopic stenting. In addition, subgroup analysis showed that surgical treatment was associated with a lower incidence of endocrine insufficiency.

Several procedures had been described as a surgical treatment for chronic pancreatitis associated pain. It can be broadly categorized into the drainage type and the resection type. Partington operation—an example of the drainage type—is an ideal one for patients with stricture predominant disease. Regarding the resection type, pancreaticoduodenectomy—pylorus-preserving or not—is a procedure of choice for mass forming chronic pancreatitis in which malignancy cannot be excluded. While Frey’s or Beger’s duodenum-preserving pancreatic head resection can be used to treat pancreatic head mass, multiple strictures and dilated pancreatic duct simultaneously. Choice of procedure depends on the endocrine and exocrine statuses of the patient, the radiological morphology of the pancreas, and the expertise of the surgeon. However, significant difference in the effectiveness in pain relief between different surgical procedures has not been seen, as demonstrated by a recent meta-analysis of RCTs [31].

Several RCTs had consistently reported that surgical drainage might be a more effective means in treating pain associated with obstructive chronic pancreatitis, authorities remained conservative regarding its recommendation [32,33]. The rationale behind is likely related to the concerns of outcome uncertainty and complications associated with an ultra-major operation. To address the former issue, the primary endpoint of this meta-analysis demonstrated significant advantage in pain relief with surgical intervention [OR 0.33, 95% CI 0.23–0.47, *p* < 0.001, I^2^ = 4%]. As supplemented by the Funnel plot and sensitivity analysis, the chance of confounding effect from publication bias (Figure 8) and study homogeneity (Figure 9) is low. Regarding latter concern of complication rate, five out of the seven included papers provided data for comparison. Significant difference between two treatment modalities was not demonstrated; however, interpretation should be done with caution given the relatively high I^2^ value (I^2^ = 49%). Head-to-head comparison for surgical and endoscopic complications risked oversimplification due to heterogeneity in the definitions and magnitude of seriousness of various complications. It should be noted that complications after pancreatic surgery and after endoscopic procedures are not equal regarding to fatality and risk of severe morbidity.

While the optimal timing of surgery has not been established, these RCTs seemed to suggest surgical intervention as a promising option regardless of time. Ali et al. [34] suggested offering patients surgery within three years of symptomatic chronic pancreatitis as more pain relief [OR 1.8, 95% CI 1.0–3.4, *p* = 0.03] and less endocrine pancreatic insufficiency [OR 0.57, 95% CI 0.33–0.96, *p* = 0.04] were observed in these patients. However, Boerma et al. [35] reported that endoscopic stenting of the pancreatic duct did not affect subsequent pancreaticojejunostomy in patients with late stage of chronic pancreatitis. As our result has shown a greater likelihood of pain relief with surgical intervention, it should be considered as soon as the endoscopic approach has failed to achieve significant improvement.

Another point to note is that the age of the patients in this meta-analysis tended to be younger in the surgery group. In the study by Issa et al. [21], the median age was seven years younger in the surgical group; in the study by Cahen et al. [22], the median age was 46 years in the surgical group and 52 years in the endoscopic group [*p* = 0.07]. It is understandable that younger patients were more likely to be offered surgery given their better physiological reserve for surgical trauma. However, the potential confounding effect of age difference should not be overlooked. Higher rate of endocrine insufficiency [OR 2.10, 95% CI 1.20–3.67, *p* = 0.01] in the endoscopy group in this meta-analysis could be a result of the natural aging process or a “burnt-out” pancreas damaged by prolonged inflammation.

Regarding the surgical approach, all studies had a certain proportion of patients who received surgery with a resection component (pancreaticoduodenectomy or Frey’s procedure). For instance, 41.5% (17/41) and 60% (61/76) of the surgical operations in the studies by Issa et al. [21] and Dite et al. [23], respectively included pancreatic resection. It is generally believed that endoscopic drainage is less effective in dealing with pain from an inflammatory pancreatic mass. As such, inclusion of such a patient population might inherently be a bias against the endoscopic approach.

Concerning secondary outcomes, it would be challenging to draw any direct conclusions due to the heterogeneous population definitions and outcome measurements. This is particularly true for length of hospitalization. Although the two approaches resulted in statistically similar duration of hospitalization [OR −0.54, 95% CI −1.23–0.15, *p* = 0.13, I^2^ = 87%], the high I^2^ value suggested the existence of uncertainty in the pooled result due to heterogeneity. 

There were a few limitations in this meta-analysis. First of all, due to heterogeneity of the definition of pain relief and pain onset in patients with painful obstructive chronic pancreatitis in this meta-analysis, further study with standardized protocol may be required to validate our results. Secondly, only three RCTs were included and they were conducted in different clinical situations; although we tried to contact with authors in each literatures, there was also paucity in data on gender, age, co-morbidities, smoking, alcohol use, and the type of endoscopic therapy in the studies (i.e., ESWL before ERCP, number of ERCPs, success rates, caliber and types of stents, repeat ESWL) in the dataset that cannot be examined in a meta-regression model, which limits the quality of results this might weaken the power of this meta-analysis. Last but not least, recent development in laparoscopic/robotic pancreas surgery and advances in endoscopic ultrasound pancreatic intervention were not considered in the current analysis, future studies focusing on these aspects are awaited. 

In summary, surgery is a promising option to achieve overall pain relief in chronic pancreatitis due to ductal obstruction, with possibly better glycemic profiles in the long run when compared to endoscopic drainage. However, considering the invasiveness of surgery, no significant differences in complete or partial pain relief, and heterogeneity of a few parameters between two groups in our SRMA, endoscopic drainage may be still preferred over surgical drainage as an initial attempt or in patients who are unsuitable for surgery and surgical drainage may be considered when endoscopic drainage fails. A sensible treatment algorithm for patients with painful pancreatitis could be a “step-up” approach with endoscopic treatment first, followed by surgery in the non-responders.

## Figures and Tables

**Figure 1 jcm-10-02636-f001:**
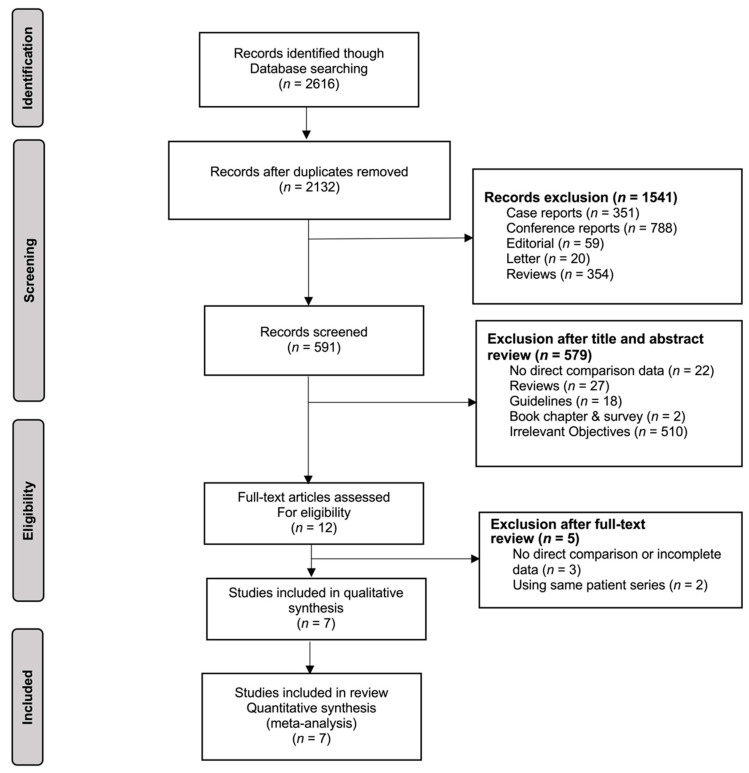
Flowchart illustration of the study search method, inclusion, and exclusion of the included studies.

**Figure 2 jcm-10-02636-f002:**
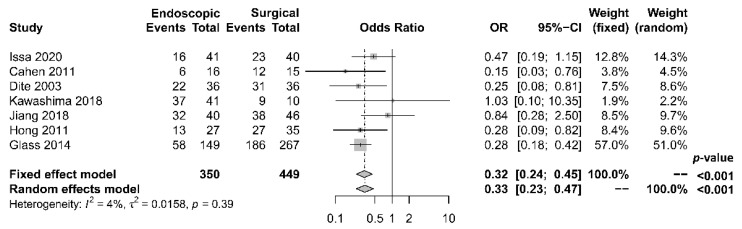
Forrest plot of the effect of endoscopy and surgery on overall (complete and partial) pain relief.

**Figure 3 jcm-10-02636-f003:**
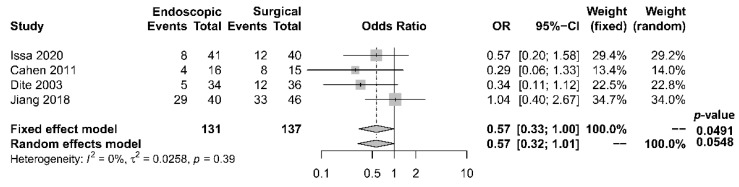
Forrest plot of the effect of endoscopy and surgery on complete pain relief.

**Figure 4 jcm-10-02636-f004:**
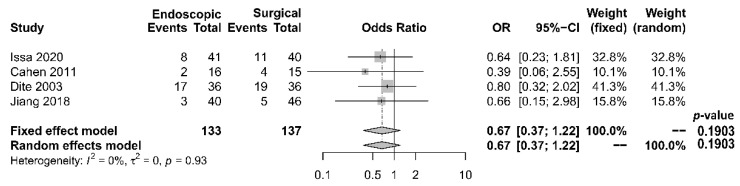
Forrest plot of the effect of endoscopy and surgery on partial pain relief.

**Figure 5 jcm-10-02636-f005:**
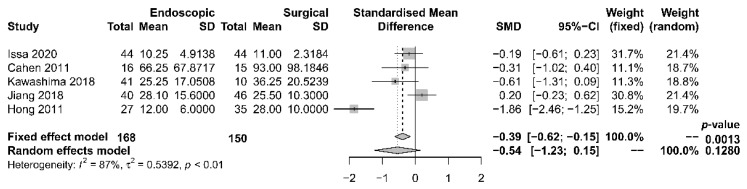
Forrest plot of the effect of endoscopy and surgery on length of hospital stay.

**Figure 6 jcm-10-02636-f006:**
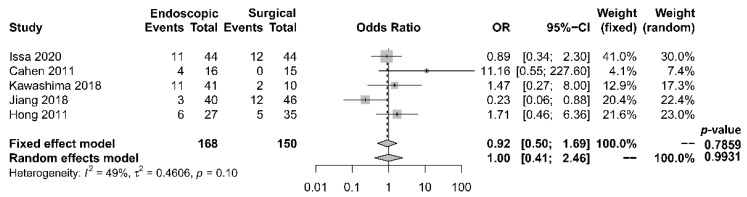
Forrest plot of the effect of endoscopy and surgery on complication rate.

**Figure 7 jcm-10-02636-f007:**
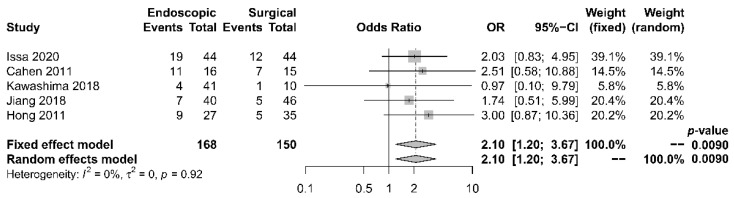
Forrest plot of the effect of endoscopy and surgery on endocrine insufficiency.

**Figure 8 jcm-10-02636-f008:**
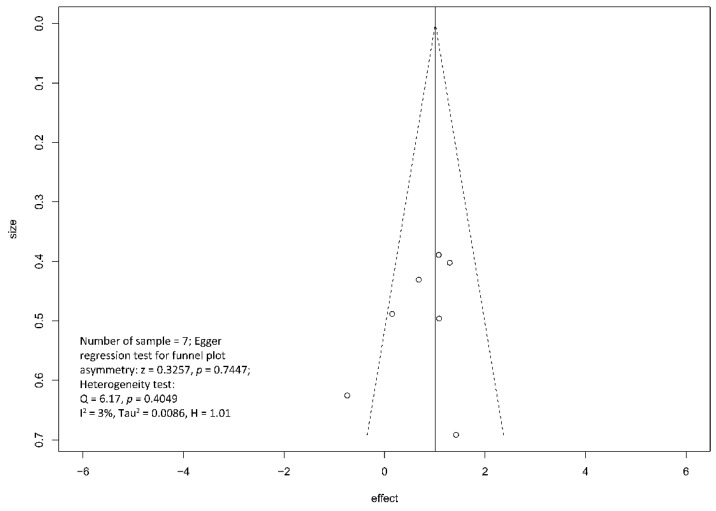
Funnel plot for the assessment of the presence of publication bias for pain relief (complete and partial) meta-analysis.

**Figure 9 jcm-10-02636-f009:**
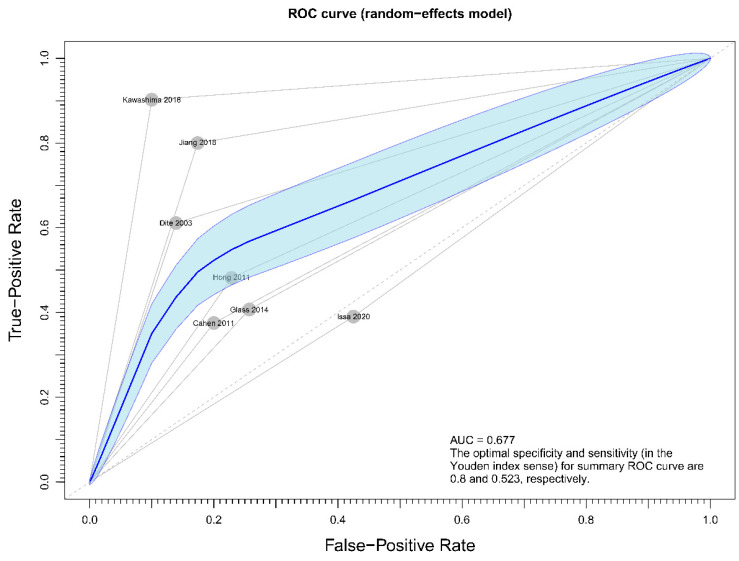
Figure showing sensitivity analysis on primary outcome (overall pain relief).

**Table 1 jcm-10-02636-t001:** Characteristics of included studies.

Study	Country	Method of Pain Assessment	Study Group	Number of Patient	Complete and Partial Pain Relief (%)	Complete Response (%)	Partial Response (%)	Hospital Stay (days)	Complication Rate (%)	Mortality Rate (%)	Definition of Exocrine Insufficiency	Exocrine Insufficiency (%)	Definition of Endocrine Insufficiency	Endocrine Insufficiency (%)	Total Cost (USD)
Issa et al. (2020)	Netherlands	IPS	Endoscopy	44	39	20	20	10	25	0	Fecal elastase < 200 μg/g	90	Need for glycemic control treatment	43	-
			Surgery	44	58	30	28	11	27	0		93		27	-
Cahen et al. (2011)	Netherlands	IPS	Endoscopy	16	38	25	13	13	25	6	Fecal elastase < 200 μg/g	100	Need for glycemic control treatment	69	31,048
			Surgery	15	80	53	27	11	0	0		87		47	25,042
Dite et al. (2003)	Czech Republic	Melzack	Endoscopy	36	61	15	46	-	-	-	-	-	-	-	-
			Surgery	36	86	34	52	-	-	-		-		-	-
Kawashima et al. (2018)	Japan	Doses of analgesics	Endoscopy	41	100	-	-	18	27	0	PFD < 70%	10	HbA1c >6.1%	10	19,908
			Surgery	10	100	-	-	23	20	0		10		10	20,964
Jiang et al. (2018)	China	-	Endoscopy	40	80	73	8	28	8	0	New onset of steatorrhea	13	New onset of diabetes mellitus	18	1317
			Surgery	46	83	72	11	26	26	0		15		11	1980
Hong et al. (2011)	China	IPS	Endoscopy	27	47	-	-	12	22	0	-	54	Any of new onset or deterioration of diabetes mellitus, need for glycemic control treatment	35	-
			Surgery	35	77	-	-	28	14	3		29		13	-
Glass et al. (2014)	United States	-	Endoscopy	145	41	-	-	-	-	-	-	60	-	-	-
			Surgery	35	74	-	-	-	-	-		33		-	-

IPS: Integrated pain score; PFD: Pancreatic functioning diagnosant HbA1c: Hemoglobin A1C.

**Table 2 jcm-10-02636-t002:** Quality assessment for the included studies.

Study	Design	Method of Grading	Jadad Scale						Newcastle–Ottawa Score			
			Randomization	Blinding	Withdrawal and Dropout	Method of Randomization	Method of Blinding	Total	Selection	Comparability	Outcome	Total
Issa et al. (2020)	RCT	Jadad Scale	1	0	1	1	0	3	-	-	-	-
Cahen et al. (2011)	RCT	Jadad Scale	1	0	1	1	0	3	-	-	-	-
Dite et al. (2003)	RCT	Jadad Scale	1	0	0	1	0	2	-	-	-	-
Kawashima et al. (2018)	Retrospective, comparative cohort study	Newcastle–Ottawa Score	-	-	-	-	-	-	2	2	2	6
Jiang et al. (2018)	Retrospective, comparative cohort study	Newcastle–Ottawa Score	-	-	-	-	-	-	4	2	3	9
Hong et al. (2011)	Retrospective, comparative cohort study	Newcastle–Ottawa Score	-	-	-	-	-	-	4	2	2	8
Glass et al. (2014)	Retrospective, comparative cohort study	Newcastle–Ottawa Score	-	-	-	-	-	-	3	1	2	6

## Data Availability

Data is contained within the article.

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
