# Peer review of "Endoscopic versus Surgical Intervention for Painful Obstructive Chronic Pancreatitis: A Systematic Review and Meta-Analysis"

_jcm, 2021, doi:10.3390/jcm10122636_

Round 1

Reviewer 1 Report

"Endoscopic versus Surgical Intervention for Painful  Obstructive Chronic Pancreatitis: A Systematic Review and  Meta-Analysis" is a well performed meta-analysis on a very interesting clinical question. The manuscript is well written and dealing with an interesting and clinical relevant question.

The manuscript is well organized and conclusions are supported by results. I have only minor comments.

In figure 1. Records exclusion n= 1541 does not equal the sum of the reported specific exclusion reasons (Case reports, Conference reports etc). For easier readability the main headings in the boxes (Record exclusions, Exclusion after title and abstract review and Exclusion after full-text review) could be in bold or similar indicating this to be a header and the reported variables below as subgroups of the header.

Although heterogeneity is discussed specifically related to complications, it should be noted in discussed that complications after pancreatic surgery and after endoscopic procedures are not equal regarding to fatality and risk of severe morbidity.  As discussed, this could not be measured as mortality or length of hospital stay in this meta- analysis but these are still relevant argument for choosing endoscopic treatment for some patients in the clinical situation.

Author Response

Dear reviewer, thank you very much for your kind comments. The following amendments made according to your suggestions:

  1. number typo and new formatting in Figure 1. were made
  2. difference in nature of the complications between surgical and endoscopic treatment was stressed in the discussion part

Reviewer 2 Report

The Review by Ka Wing Ma et al gives a comprehensive overview of current approaches to painful obstructive chronic pancreatitis (CP) while trying to provide an Meta-Analysis of both randomized and retrospective studies addressing comparison between endoscopic and surgical treatment options.

The text is written in correct and readable English, with a logical composition. The authors have performed appropriate statistical methods considering the heterogeneity of data within the studies included. The authors are well aware of numerous limitations within their study while nicely pointing out important aspects and interpretations of the current data. In my opinion the work is an important contribution to the field.

However, there are several issuses that require further rectification:

Introduction

  1. While focusing on therapeutical options for CP the authors only address  etiology at a short glance. On the one hand this is justifiable because obstructive CP represents a common final path of various underlying pancreatic diseases. On the other hand I would strongly recommend to give a more comprehensive overview of different disease causes as they might also influence a phycisians decision-making (e.g. a young patient with hereditary CP might be more favorable for surgical interventions compared to a patient with nutritive-toxic CP).
  2. Pain is mainly caused by ductal obstruction, but I would also mention repetitive inflammatory activity as a reason for pain (also CP patients without ductal obstruction report repetitive painful interceptions)

Methods:

  1. Did you include studies that combined shockwave lithotripsy and endoscopy? Please clarify.

Discussion:

  1. The whole Text is built upon as an "either-or" discussion wich -to my own experience- does not precisely reflect an appropriate algorithm in treatment of patients with cP. I would kindly suggest to provide a more differentiated view: Like in many sub-fields of Gastroenterology/Endoscopy surgery should be unterstood as an complementary method. For many patient a step-up approach might be an appropriate way to prevent over-treatment. For other patients predictors like long pain history, high pain intensity or present endocrine insufficiency might suggest an early surgical approach. Therefore I recommend (i) to better discuss predictors for necessity of surgical treatment and (ii) to discuss the option of step-up approaches (surgical treatment for non-responders of endoscopy).

Author Response

Thank you very much for your kind comments, amendment have been made according to your suggestions as follows:

  1. Other causes of chronic pancreatitis have been supplemented in the introduction part
  2. repetitive parenchymal inflammation has been supplemented as one of the causes of chronic pain in chronic pancreatitis
  3. studies which used shockwave lithotripsy and endoscopy as a treatment modality were excluded. This point was clearly supplemented in the method part of the revised manuscript 
  4. We suggested a possible treatment algorithm for chronic pancreatitis as the reviewer suggested at the end of the discussion. We agree that "step-up" approach is a sensible way to go, judging from the invasiveness of surgical treatment. However, we hope the reviewer could accept our limitation that, we could not make in-depth or clear cut recommendation regarding which patient group should receive what kind (surgical or endoscopic) treatment base on the results from this current meta-analysis.